# Codes of Conduct at Zoos: A Case Study of the Chengdu Research Base of Giant Panda Breeding

**David Fennell [1] and Yulei Guo [2,\*]**

1   Geography and Tourism Studies, Brock University, St. Catharines, ON L2S 3A1, Canada; dfennell@brocku.ca
2   Tourism Department, Chengdu Research Base of Giant Panda Breeding, Chengdu 610081, China
\*   Correspondence: yulei.guo@panda.org.cn

**Abstract:** Zoos consistently implement codes of conduct in efforts to manage visitor behaviour. However, few studies have examined the use of the codes of conduct in zoos, even though they carry significant ethical implications regarding the relationship between humans and animals in society. This study provides an explorative investigation into the use of codes of conduct at the Chengdu Research Base of Giant Panda Breeding (Panda Base). Positioning the Panda Base as a place to negotiate the boundaries between humans and animals, this study surveyed visitors' initial engagement with the Base's code of conduct, their compliance with the code, and their assessment of the code. The findings point to a significant disparity between how visitors engage with and perceive the value of the code, which failed to prevent visitors from having close contact with animals at the Panda Base. We argue that Foucault's philosophy on taboos in modern society can help us understand the ineffectiveness of the codes of conduct in zoos. However, Kant's philosophy can orient human-animal interactions more ethically and provide an opportunity to consider the significance of codes of conduct in zoos. Suggestions for improving the effectiveness of codes of conduct at zoos are provided.

**Keywords:** codes of conduct; zoos; animal-human interaction; giant panda

## 1. Introduction

Although formally places of entertainment, modern zoos have extended their mandate to include activities deemed more socially and ecologically responsible, including care for the welfare of exhibited animals, educating and engaging the public, conserving species/habitats, and conducting academic research on animals [1,2]. However, studies consistently indicate that entertainment is one of the most important objectives of zoos [3–6], based on an estimated 700 million people who visit zoos and aquariums annually [7,8]. Zoos are big business. Studies over several years have also shown that visitors who mainly seek entertainment can have significant negative effects (the "visitor effect", e.g., animal behavioural and physiological change) on the welfare of animals at zoos due to their lack of concern for the co-presence of animals [9]. These effects are taking place alongside a global movement concerned with the welfare of animals used in several different sectors, e.g., agriculture [10]. Mitigating the negative impacts of visitor effects is becoming a need that zoos worldwide are attempting to address.

Tourism scholars have proposed the term "zoo tourism" to more explicitly understand the conflicts that may exist between the educational, scientific, and entertainment roles of zoos [11]. Studies have demonstrated that zoo tourism provides opportunities for biodiversity conservation [12,13], conservation education [5,14,15], and economic benefits to locals [16]. According to Mason [11], zoos have the potential to become ecotourism attractions and contribute to a sustainable future of tourism. Hence, for tourism scholars, the mitigation of negative visitor effects can be an approach to sustainable tourism development.

A tool now being used liberally in zoos, globally, to mitigate the visitor effect is codes of conduct (codes of conduct govern actions, while codes of conduct govern decision-making). Codes of conduct are now a fixture in zoos for the purpose of managing behaviour, often articulated within the broader context of compassionate conservation which specifically addresses the individual welfare and wellbeing of animals [17–21]. However, few studies have examined the use of codes of conduct in zoos, even though codes carry significant ethical implications regarding the relationship between humans and animals in contemporary society. According to Malloy and Fennell [22], codes of ethics in tourism serve as a vehicle for communicating an organization's ethical culture to employees, visitors, and other stakeholders. While the zoo is an essential modern institution for managing the relationship between humans and animals [23], codes of conduct in zoos are a manifestation of this relationship and a way of communicating organizational messages to visitors.

Competing demands and priorities between entertainment and education, welfare, and conservation suggest a chasm that zoo codes of conduct must bridge. Zoo codes of conduct must specify visitor obligations and responsibilities in order to achieve conservation and education objectives. In an effort to shed light on these conflicting aims of zoos, the purpose of this paper is to investigate zoo codes of conduct in order to establish a fundamental understanding of their use, ethical value, and compliance among visitors. Specifically, through the use of Reidenbach and Robin's [24,25] multidimensional ethics scale, which has been used in other related contexts in tourism studies research, we investigate (1) visitors' acceptance of codes of conduct, (2) whether tourists use the codes when booking tickets online and how they interact with the guideline window, and (3) tourists' evaluation of the importance of the codes. The Chengdu Research Base of Giant Panda Breeding provided an empirical case study for this paper. Theoretically, we first explore zoos as places for ethical consideration through Foucauldian and Kantian philosophy and further discuss the effectiveness of codes of conduct in serving zoos' ethical challenges.

## 2. Literature Review

This section introduces the ethical debates around the zoo as a modern institution between human beings and animals to demonstrate the oppression and silence that animals experience in animal organizations. Acknowledging this unequal animal-human relationship suggests that codes of conduct either serve to maintain or reinforce this existing relationship in contemporary society. It challenges us to think of a code of conduct that can tip this imbalance.

### 2.1. Zoos as Places for Ethical Consideration

Several studies have examined zoos from the perspective of the Foucauldian tradition [26–31]. Situated within the framework of Foucault's concepts, such as gaze, biopolitics, power, and panopticon, zoos emerge as contemporary establishments where human civilization extends its governance and biotechnological practices to encompass nonhuman beings. Because of the fluid boundary between humans and nonhumans, Braverman [30] notes that how zoos manage and conserve animals mirrors the existence of human beings in modern society. Braverman's [30] view allows zoogoers to assume a particular role: as visitors observing animals, these individuals can adopt a vantage point to critically examine the social institutions that have shaped and regulated human life within contemporary society. The ability to look at animals as being different from humans has also been investigated through the concept of the tourist gaze, where tourists gain privilege over the objects of their curiosity [32].

On this account, and in accordance with Foucault, the gaze enacts constructed regimes of power giving licence for human domination and control over animals. The dynamics of visitor-animal interactions are thus shaped by the intricate web of social relations constructed by human society. In this context, animals are often relegated to being passive and voiceless objects, existing primarily for human observation and scrutiny. As such, an animal's existence, its voicelessness, is no broader than the network of relations in

which they emerge as observed, preserved, and studied [33]. In Foucault's framework, the transformation of zoos from an organization that historically provided entertainment to a place dedicated to animal conservation and education [3–6] does not fundamentally undo the power structures relegating animals as subaltern others.

Acknowledging the unequal power relationship between visitors and animals that zoos institutionalize reinforces the need to consider potential ethical relationships between humans and animals. Fennell [34] suggests that captive animal venues, and their visitors, can transition from "constructed care" to a care ethic that flows between species. As suggested by its name, constructed care refers to social relationships shaped, in part, by the tourist gaze that dictates how visitors consume captive animal products at zoo venues, even when presented with discourses that emphasize empathy towards captive animals. Constructed care is defined as the adoption of a pathos that seeks to impose its legitimacy on others, like tourists, whilst being embedded in an ethos framed by an institutional structure that is instrumental and utilitarian by nature [34]. In such cases, we legitimize trade-offs which reinforce the value of pleasure and profit at these venues at the expense of animals [35,36]. By contrast, an ethic of care differentiates itself from constructed care by establishing an ethical foundation for interactions between animals and visitors [21].

We argue that this ethical foundation in zoo venues can be aided by Immanuel Kant's concept of a "thing in itself". Kant holds that things should have a status independent of representation and observation. We tend to know things by their appearance, which in turn is determined by how our senses interpret these things, which in contrast are often unknowable and unexperienceable. Thomas Nagel [37] furthered this idea by suggesting that we are confined by the limitations of our own mind in attempting to interpret—to actually know—what it is like to be another species, such as a bat.

Kant also proposed the concept of "purposefulness with a purpose" as a response to this indeterminate uncertainty. According to Zuckert [38] (p. 81), purposiveness without a purpose "characterizes an object that seems useful for a purpose (that we can only 'grasp' as such), but we do not know for which purpose, do not claim that it is in fact useful". Looking at zoo animals through Kant's lens, they become beings deeper and more meaningful than the social institution from which they emerge.

Expanding on the idea of purposiveness and thing in itself, Kant, in Critique of Judgement [39], advocated using reflective teleological judgement to understand the relationship between nature and human beings. The teleological judgement acknowledges that how an object appears to be itself can be a consequence of being a "thing in itself" and how it is represented and constructed in social relations. Kant hypothesized that, like human morality, purposiveness also endeavours for the highest good. Kant concludes, based on his belief in nature's beneficent purpose, that human morality is an integral element of nature's teleology. It is not a coincidence that Malloy and Fennell [22] emphasized the importance of using teleology as the ethical approach to guide tourists' actions. These authors found that a teleological strategy could guide visitors' conduct more effectively because it stresses the morally sound outcomes of actions. Referring back to Kant's argument, the teleological approach to conceiving the codes of conduct is not merely necessary but mandatory.

In his object-oriented ontology (OOO), Harman [33] (p. 251) proposes a similar idea to Kant's, suggesting that an inanimate object is "deeper than all relations". For these scholars, zoo animals should be "animals in themselves" whose existence and connection occur beyond constructed care and gaze because they have a very special kind of intrinsic value. Harman [33] argues in OOO that the object is deeper than its social relationships and could never reveal itself to us, echoing Nagel's [37] views on the inability of humans to understand the nature of animal others. As such, rather than looking at zoos as institutions where animals' lives have been politicized and manipulated for human interest, Kantian philosophy points out that zoos can be places for ethical conduct if animals are respected as "things in themselves".

## 2.2. The Effectiveness of the Codes of Conduct and Tourism

The prevalence of visitor codes of conduct in zoos suggests that an assessment of their effectiveness should be a priority as part of their implementation, a topic that has received considerable attention in the broader literature. Doig and Wilson [40], for example, suggested that there needs to be more evidence of the effectiveness of corporate codes of conduct, a conclusion echoed by Yallop [41] over a decade later. Similar conclusions were made by Valentine and Barnett [42], who found that there is difficulty reaching consensus on how valuable and effective codes of ethics are. Kaptein and Schwartz [43] reviewed 79 empirical studies addressing the effectiveness of codes of conduct and showed that scholars have divergent and even conflicting views on their value. Babri et al. [44] point out that existing studies on code effectiveness are fragmented because the concepts and variables employed are different between studies. For Stevens [45], what makes codes of conduct effective is a question more important than whether the codes have an effect.

Several studies have explored the use of codes of conduct in tourism since the late 1990s [22,46]. Malloy and Fennell [22] recognized that ethical conduct has become a concern not only among tourism operators and members of tour organizations but also among tourists themselves. In their review of 414 statements of codes of conduct developed in tourism, Malloy and Fennell [22] pointed out that almost 45% of these have been developed for tourists. However, Malloy and Fennell [22] pointed out that only minor attention has been paid to the effectiveness of the codes of conduct in tourism. Fennell and Malloy [47] suggested that the success of codes depends on a good understanding of their target audience, and stress the importance of embodying a sense of respect, justice, and dignity into the value-set codes of conduct. For these authors, the most effective codes are teleological rather than deontological because they provide the rationale and justification behind the use of codes rather than imply or reinforce a desired conduct. A central aim of codes is to act as a communication device for target audiences [45,48], where education is centred on prevention rather than cure [49].

In animal-based tourism, several species, or orders (e.g., cetaceans), have been the target of codes of conduct [50–58]. While many of these studies have focused on the content of codes of conduct, code effectiveness has been investigated on the basis of tourist compliance, for example, on whether tourists have maintained the advocated distance (2 m) from the whale shark [57] and visitors' adherence to the codes of conduct [55,58] through on-site observations. According to Smith, Scarr, and Scarpaci [58], more work is needed to address visitor compliance and animal behaviours when investigating human-wildlife interactions.

Codes of conduct are now an established feature of captive animal venues, where managing visitor use (in large numbers, as noted above) is paramount in balancing this use with animal welfare. The World Association of Zoos and Aquariums [59] developed the "Guidelines for Animal-Visitor Interactions", which stemmed from its own code of conduct and animal welfare [60], World Zoo and Aquarium Animal Welfare Strategy [61], and WAZA resolution on animal interactions [62]. This code of conduct is based on six primary recommendations:

1. Avoid having animals in any interactive experience that would compromise their welfare.
2. Animals involved in direct contact situations should receive appropriate training for visitor interactions in order to reduce potential discomfort or stress responses.
3. Make no unnecessary demands on animals and ensure that visitors do not provoke or create discomfort or stress responses in the animals.
4. Provide animals with a choice of whether to participate or not in the interactions. Allow adequate rest time and ensure that an animal displaying any indication that it does not want to participate is immediately removed from the interactive experience.
5. All walk-through habitats, touch pools, and petting areas/touch paddocks where animals are in close proximity to visitors should be of a suitable size to provide for species-appropriate needs and have suitable refuge areas for the animals.

6.  Any feeding during an interaction must be regulated so it is consistent with the animal's overall appropriate diet and health care. This food must not be the only access to food or the whole diet for the animal and the animal must have a choice whether to accept this food.

However, as illustrated by Learmonth [63], "individual institutional adherence to these "guidelines" in varying regions may be incomplete, inadequate, or altogether ignored (in favour of financial viability or human experience, for example)" (pp. 5–6). A comprehensive report by World Animal Protection [64] shows that even among gold-standard zoos around the world, some of which are WAZA members, mistreatment of animals takes place through visitor-animal interactions such as selfies, petting, and feeding to make more money [65]. These findings provide traction to Fennell's [34] belief that captive animal venues, even though they may be accredited or certified, practise constructed care rather than an ethic of care.

In sum, there is consensus over the need for more research on the effectiveness of codes of conduct. Existing studies provide a fragmented picture of the use of codes in business because of the use of varying definitions of key terms, data, methodological deficiencies, and a need for explicit theory. In tourism, scholars have suggested that attention to the stakeholders' needs and education can be the key to the success of codes of conduct. In contemporary society, zoos are places where people negotiate the borderline between animals and human beings. Zoo visitors, wittingly or unwittingly, are participants in this ongoing negotiation. Codes of conduct in zoos are one of the manifestations of this ethical consideration. Hence, researchers have also developed metrics for evaluating animal-tourist interactions based on tourists' compliance with the established codes of conduct.

## 3. Method

### 3.1. Panda Base's "Visitor Guideline"

With more than 200 giant pandas on display daily, the Panda Base is the largest giant panda breeding institution in the world (the location of the Panda Base; see Figure 1).

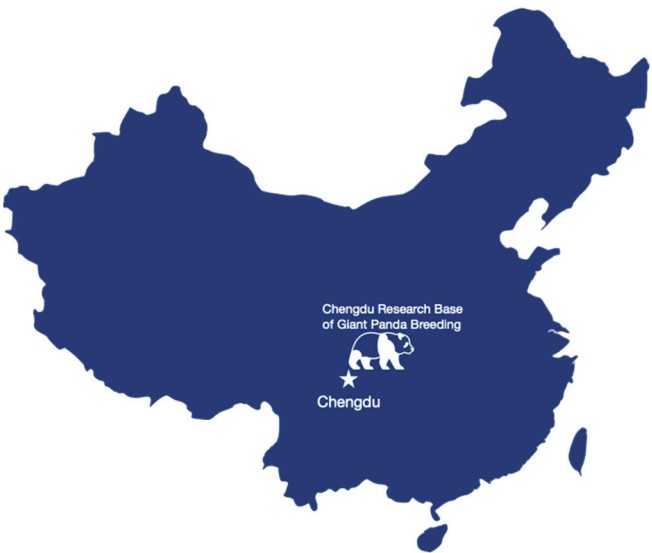

**Figure 1.** The location of Chengdu Research Base of Giant Panda Breeding.

Before the outbreak of COVID-19, more than 9 million visitors travelled to the Panda Base to encounter giant pandas in 2019. Despite being a breeding facility unique to its kind, the Panda Base is characterized as a zoo as a function of its mission and management practices.

Like many other Chinese zoos, the Panda Base relies on its Visitor Guidelines to influence the conduct of visitors. The Panda Base Visitor Guidelines (see Appendix A) is

published as a part of the ticketing window, which pops up as tourists book their visits online. The Visitor Guidelines are, hence, a mandatory read for visitors planning to see the pandas. Following Malloy and Fennell [22], we note that the Visitor Guidelines make minimum use of teleological statements, which can be found only at the beginning and the end of the guidelines. These statements focus on how the Guidelines can better the lives of animals at the Panda Base. The rest of the Visitor Guidelines employ a deontological approach that bans visitors from certain types of behaviour. As such, the Guidelines contain little educational information to help visitors understand the consequences of their behaviour. Furthermore, the Guidelines are anthropocentric—they do not address pandas and other animals at the Panda Base as subjective beings of their own ends. Instead, the Guidelines recognize that visitors interact with animals through teasing, feeding, touching, and noisemaking and attempt to control these depreciative behaviours.

Based on the literature review, a visitor code of conduct, such as the Panda Base Visitor Guidelines, can be less effective in regulating the conduct of visitors because their deontological approach fails to build a respectful bond between animals and visitors. In this study, we investigate (1) visitors' acceptance of the Visitor Guidelines, (2) whether tourists noticed the Visitor Guidelines when booking online and how they interacted with the guideline window, and (3) tourists' evaluation of the importance of the Visitor Guidelines on a 6-point Likert Scale (1 = Extremely unnecessary; 6 = Extremely necessary).

*3.2. Scenarios and Measurement*

This paper replicates the Multidimensional Ethics Scale (MES) that Fennell and Malloy [66] adopted from Reidenbach and Robin [24,25] to measure the ethical nature of tourism operators. The previous studies established validity and credibility that can support an explorative study like this. MES provides a semantic differential scale representing three dimensions of ethical behaviour—deontological, justice, and relativistic theories. The deontological dimension evaluates one's obligation to abide by rules, contracts, and duties. The justice dimension reflects the cultivation of the value of fairness, goodness, justice, and rightness in an individual's early training by family and religion. The relativistic construct refers to the particular sociocultural context in which the individual acts. The three dimensions are reflected in eight scale items.

The MES has been used extensively in tourism studies [67–72]. In order to capture the participants' responses more accurately in this study, new Panda Base-specific visitor scenarios were developed by the researchers. The three scenarios are: (1) Intimate contact (less than 3 m) with free-roaming and wild animals; (2) Smoking; and (3) Trampling on the lawn. Although all three scenarios are misconducts strictly banned by the Visitor Guidelines, one of the researchers working at the Panda Base has witnessed tourists breaking all three scenarios frequently. Stopping visitors from committing the three misbehaviours has been a management routine of Panda Base employees and sometimes these tasks are met with obstacles (e.g., visitors refusing to stop smoking or stepping off the lawn). All three scenarios have signposts built visibly at the Panda Base. Different from the scenarios constructed with pseudo characters by Fennell and Malloy [66], the empirical context of this study allowed the researchers to investigate whether tourists misbehaved at the Panda Base before taking the survey (A). Section A is a single-choice question aiming to streamline participant responses and ensure clarity in data interpretation. For each scenario, we ask tourists to identify their experiences with the misconduct. Combined with MES (B), the three scenarios are as follows:

Scenario 1:

1.  Having intimate contact (<3 m) with free-roaming and wildlife animals at the Panda Base

    a.  I did it.
    b.  I saw someone do it.
    c.  I stopped someone touching.
    d.  I did not see it happen.

2.    Your response to the intimate contact with animals is that it is...

| Unfair | 1 | 2 | 3 | 4 | 5 | 6 | 7 | Fair |
|---|---|---|---|---|---|---|---|---|
| Unjust | 1 | 2 | 3 | 4 | 5 | 6 | 7 | Just |
| Not morally right | 1 | 2 | 3 | 4 | 5 | 6 | 7 | Morally right |
| Unacceptable to my family | 1 | 2 | 3 | 4 | 5 | 6 | 7 | Acceptable to my family |
| Traditionally unacceptable | 1 | 2 | 3 | 4 | 5 | 6 | 7 | Traditionally acceptable |
| Culturally unacceptable | 1 | 2 | 3 | 4 | 5 | 6 | 7 | Culturally acceptable |
| Does not violate an unspoken promise | 1 | 2 | 3 | 4 | 5 | 6 | 7 | Violates an unspoken promise |
| Does not violate an unwritten contract | 1 | 2 | 3 | 4 | 5 | 6 | 7 | Violates an unwritten contract |

Scenario 2:

A.    Smoking at Panda Base

    a.    I did it.
    b.    I saw someone do it.
    c.    I stopped someone touching.
    d.    I did not see it happen.

B.    Your response to smoking at the Panda Base is that it is...

The MES scale

Scenario 3:

A.    Trampling on the lawn at the Panda Base:

    a.    I did it.
    b.    I saw someone do it.
    c.    I stopped someone touching.
    d.    I did not see it happen.

B.    Your response to trampling on the lawn at the Panda Base is that it is...

The MES scale

In total, 852 visitors in May 2023 completed the survey at the exit of the Panda Base. Three data entries with unrealistic age numbers were deleted from the dataset, leaving 849 valid samples. All visitors voluntarily participated in the survey and received a panda magazine as compensation. According to Chinese law, an anonymous study of this nature, which does not collect personally identifiable information encompassing biometrics, religion, medical health, financial account, or location, is not subject to ethical committee approval. Figure 2 provides an overview of the methodological framework that this paper incorporates. The three analytical steps are composed of measurements that the questionnaire is designated to address.

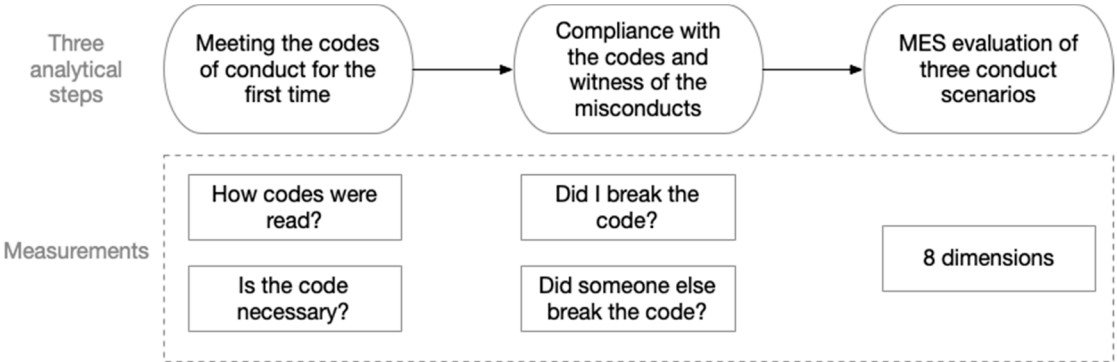

**Figure 2.** Three analytical steps that this paper takes.

## 4. Results

### 4.1. General Summary

Table 1 presents the demographic information of samples collected in the study. The summary shows that the sample collected is inconsistent with other studies performed at the Panda Base [73,74]. We observed no significant fluctuation in demographic features between datasets collected for other studies based on a similar data collection procedure.

**Table 1.** Demographic description of samples collected for this study.

| Measure | N | % | Reading Styles of the Guidelines [1] | | Importance of the Guidelines [2] | |
|---|---|---|---|---|---|---|
| | | | **Mean** | *p* | **Mean** | *p* |
| Gender | | | | | | |
| Female | 521 | 61.1 | 1.45 | 0.681 | 5.12 | 0.011 * |
| Male | 328 | 38.6 | 1.48 | | 4.86 | |
| Age | | | | | | |
| Under 18 | 46 | 5.4 | 1.67 | | 5.13 | |
| 18–24 years old | 225 | 26.5 | 1.44 | | 4.81 | |
| 25–34 years old | 323 | 38.0 | 1.51 | 0.012 * | 5.09 | 0.008 * |
| 35–44 years old | 162 | 19.1 | 1.31 | | 4.68 | |
| 45 and over | 93 | 11.0 | 1.48 | | 5.50 | |
| Educational background | | | | | | |
| Junior high school | 81 | 9.5 | 1.59 | | 5.40 | |
| Senior high school | 148 | 17.4 | 1.39 | 0.218 | 4.99 | 0.080 |
| College/University | 310 | 36.5 | 1.47 | | 5.04 | |
| Postgraduate | 310 | 36.5 | 1.46 | | 4.92 | |
| Visiting purpose | | | | | | |
| Holiday | 606 | 71.4 | 1.51 | | 4.91 | |
| Hanging out with friends/family | 158 | 18.6 | 1.44 | | 5.26 | |
| Education | 36 | 4.2 | 1.41 | 0.738 | 5.23 | 0.264 |
| Business | 20 | 2.4 | 1.38 | | 4.94 | |
| Others | 29 | 3.4 | 1.38 | | 4.92 | |
| Origin | | | | | | |
| First tier | 157 | 18.5 | 1.43 | | 4.87 | |
| Second tier (including Chengdu) | 302 | 35.6 | 1.43 | 0.377 | 4.96 | 0.090 |
| Third tier | 390 | 45.9 | 1.50 | | 5.14 | |
| First-time Visitor | | | | | | |
| Yes | 739 | 87.0 | 1.48 | 0.065 | 5.04 | 0.381 |
| No | 110 | 13.0 | 1.35 | | 4.91 | |
| Subscribers of giant panda topics on social media | | | | | | |
| Yes | 591 | 69.6 | 1.50 | 0.033 * | 5.13 | 0.001 * |
| No | 258 | 30.4 | 1.38 | | 4.77 | |

**Table 1.** *Cont.*

| Measure | N | % | Reading Styles of the Guidelines [1] | | Importance of the Guidelines [2] | |
|---|---|---|---|---|---|---|
| | | | **Mean** | ***p*** | **Mean** | ***p*** |
| Are you a panda fan? [3] | | | | | | |
| Yes | 566 | 66.7 | 1.54 | 0.000 * | 5.33 | 0.000 * |
| No | 283 | 33.3 | 1.30 | | 4.42 | |
| Total | 849 | 100.00 | 1.50 | | 5.00 | |

Note. [1]: The reading styles of the guidelines are coded as follows: 1 = Skimming through within 1–3 s; 2 = Read selected sections carefully; 3 = Read through the guidelines carefully. [2]: The importance of the guidelines was a 6-point Likert Scale question: 1 = Extremely unnecessary, 2 = Unnecessary, 3 = Slightly unnecessary; 4 = Slightly necessary; 5 = Necessary; 6 = Extremely necessary. [3]: We categorized the panda fans through participants' own evaluation. The same approach was applied by Fennell and Guo [73]. * $p < 0.05$

A total of 836 visitors (98.5%) suggested noticing the Visitor Guidelines window when booking their tickets. We noted that 66.8% (576) of visitors skipped through the Guidelines in less than 3 s, 20.1% (171) of visitors selected a few sections to read, and 13.1% (111) of participants read the Guidelines statement by statement. Despite the majority of visitors tending to skip the Visitor Guidelines, 59.1% (502) of visitors claimed that the Visitor Guidelines were "extremely necessary", and 118 (13.9%) participants believed that the guidelines were "necessary". The average score for the necessity of the Visitor Guidelines is 5.02 "necessary".

*t*-tests and ANOVA were performed to observe whether the demographic characteristics help distinguish the reading styles and importance of the Guidelines. The results show that women (M = 5.12), in general, considered the Visitor Guidelines significantly more important ($p = 0.011$) than did males (M = 4.86). Age, subscription to social media, and identification with being a panda fan all hold significant influence over the participants' reading styles and the evaluation of the importance of the Guidelines. Subscribers of social media and panda fans read the Guidelines more carefully and attached more significance to the guidelines. Figure 3 demonstrates visitors' witness of the three scenarios during their visits with 219 (25.04%) visitors suggesting that they had intimate contact with the free-roaming and wild animals. Smoking and trampling on the lawn had a similar curve and follow-up, but less significant peaks when participants reported seeing someone else commit the misconduct during their visits.

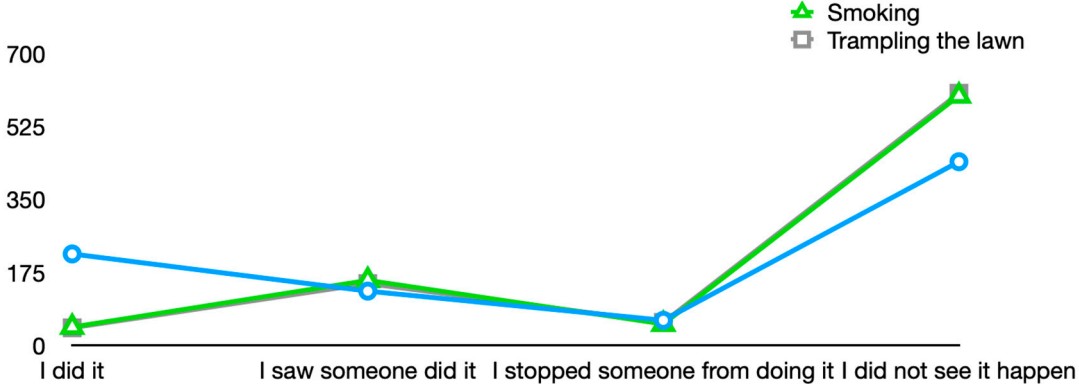

**Figure 3.** Participants' report on the three scenarios based on their visits.

### 4.2. Ethical Evaluations of the Three Scenarios

Descriptive statistics were employed to calculate mean scores for each scenario, including each item on the MES scale across scenarios and cohorts. Figure 4 demonstrates the grand means for the three scenarios investigated in this study. The higher the score, the more ethically acceptable respondents perceive a scenario. Overall, Figure 4 shows that

intimate contact with the animals, smoking, and trampling on the lawn are all considered unethical behaviours by visitors. However, intimate contact with animals is found to be more acceptable behaviour than smoking and trampling on the lawn. Figure 4 mirrors Figure 3, affirming that more tourists had intimate contact with animals because this behaviour was believed to be more acceptable. Figure 5 presents the aggregated means of the five scenarios, showing how the different items in the MES vary within a cohort. While touching the animals contributes to stable performance, we note interesting fluctuations in regard to smoking and trampling on the lawn. Specifically, participants attempted to justify the moral righteousness of smoking and trampling on the lawn, which is believed to have violated an unwritten contract.

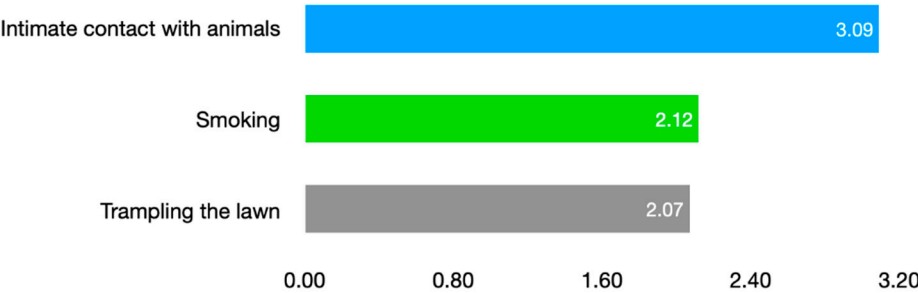

**Figure 4.** Grand means of ethical perceptions for the three scenarios.

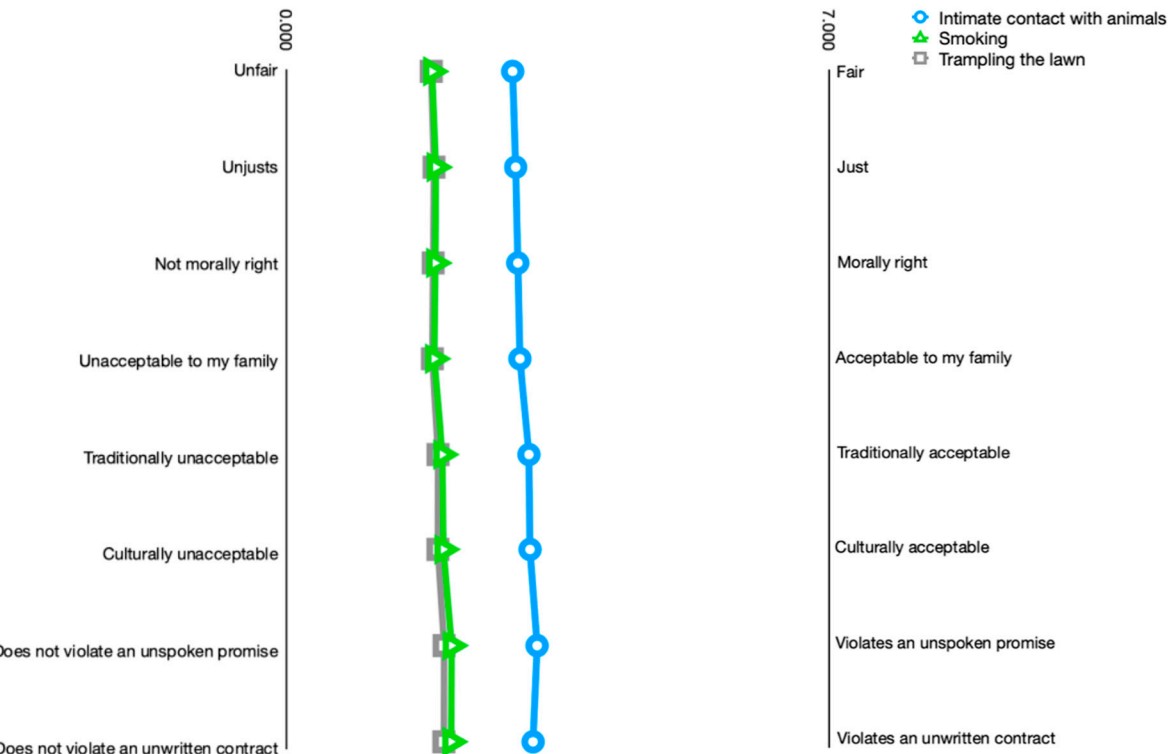

**Figure 5.** MES means aggregated by scenario.

Table 2 illustrates the ethical evaluations of the three scenarios made by three groups of readers. With ANOVA performed, it shows that the reading styles of tourists can have significant implications for the ethical evaluations of the three different scenarios. Participants who skimmed the codes of conduct in less than 3 s considered all three scenarios more ethically acceptable than did selective and thorough readers. Interestingly, we note that selective readers, rather than thorough readers, set the ethical threshold higher than thorough readers in cases of intimate contact with the animals (M = 2.89) and smoking (M = 1.74). Selective (M = 1.80) and thorough (M = 1.78) readers both have lower ethical

thresholds for trampling on the lawns at the Panda Base, yet their difference to skimming readers is still significant ($p = 0.001$).

**Table 2.** Reading styles of the guideline and the ethical evaluation of the scenarios.

| Reading Styles | N | % | Intimate Contact with the Animal | | Smoking | | Trampling the Lawn | |
|---|---|---|---|---|---|---|---|---|
| | | | Mean | p | Mean | p | Mean | p |
| I skimmed the codes of conduct in less than 3 s | 567 | 66.8 | 3.36 | | 2.28 | | 2.21 | |
| I read selected sections carefully | 171 | 20.1 | 2.89 | 0.002 | 1.74 | 0.000 | 1.80 | 0.001 |
| I read through the guideline statement by statement | 111 | 13.1 | 2.97 | | 1.88 | | 1.78 | |

Table 3 shows the perceived necessity of the codes of conduct and the ethical evaluation of the scenarios. The results show that whether the codes of conduct are perceived necessary or unnecessary does not significantly influence the ethical evaluation of the three scenarios. Interestingly, participants who considered the codes unnecessary found intimate contact with the animals (M = 0.03) and smoking (M = 2.01) less acceptable than participants who considered the codes necessary. Table 3 suggests that the ethical acceptability of different conduct can have a complex relationship with the perceived importance of the conduct guidelines.

**Table 3.** Perceived importance of the guidelines and the ethical evaluation of the scenarios.

| Are the Visitor Guidelines Necessary? [1] | n | % | Intimate Contact with the Animals | | Smoking | | Trampling on the Lawn | |
|---|---|---|---|---|---|---|---|---|
| | | | Mean | p | Mean | p | Mean | p |
| Unnecessary | 130 | 66.8 | 3.03 | 0.193 | 2.01 | 0.327 | 2.17 | 0.413 |
| Necessary | 719 | 13.1 | 3.25 | | 2.14 | | 2.05 | |

Note [1] The necessity of the Visitor Guidelines was initially measured on a 6-point Likert scale. In performing this analysis, answers are categorized into groups—unnecessary (points 1–3) and necessary (points 4–6).

Table 4 illustrates the link between participants' compliance with the guidelines and witnessing of misconduct and the ethical evaluation of the three scenarios through ANOVA analysis. The results confirm the link between tourists' highest acceptance of the conduct if they had done it in all scenarios. Specifically, participants who had intimate contact with animals believed that the conduct was slightly acceptable ethically (M = 3.95). This group also indicated a higher ethical threshold for intimate contact with the animals and accepted the behaviour. For intimate contact with animals, tourists who claimed not to see the misconduct yielded the lowest mean score (M = 2.90). In contrast, participants who saw other visitors smoking (M = 1.90) and trampling on the lawn (M = 1.86) offered the lowest ethical evaluations for the two types of conduct. Notably, participants who smoked (M = 2.77) and trampled on the lawn (M = 2.54) also believed that they had misbehaved.

**Table 4.** Participants' compliance with and witness of the misconduct and the ethical evaluation of the scenarios.

| Compliance with and Witness of the Misconduct | Intimate Contact with the Animals | | | Smoking | | | Trampling on the Lawn | | |
|---|---|---|---|---|---|---|---|---|---|
| | *n* | Mean | *p* | *n* | Mean | *p* | *n* | Mean | *p* |
| I did it | 219 | 3.95 | | 44 | 2.77 | | 42 | 2.54 | |
| I saw someone do it | 130 | 3.10 | 0.000 | 156 | 1.90 | 0.005 | 148 | 1.86 | 0.040 |
| I stopped someone doing it | 60 | 3.04 | | 51 | 2.41 | | 55 | 2.32 | |
| I did not see it happen | 440 | 2.90 | | 598 | 2.11 | | 604 | 2.07 | |

## 5. Discussion

The Visitor Guidelines of the Panda Base prohibit visitors from engaging in intimate contact with animals. However, the results show that intimate contact with animals remains the most contested ethical conduct between visitors and this animal breeding organization, with 25.8% of visitors engaging in contact behaviours. In parallel with the Guidelines' prohibition of smoking and trampling on the lawn as misconduct, visitors all ascribed lower ethical values to these two behaviours than having contact with the animals. Ballantyne, Packer, and Sutherland [75] note that touching animals makes a lasting impression on wildlife visitors. Quiros's [55] study shows that 56% of swimmers interacting with whale sharks in Donsol, Philippines, would infringe the minimum distance that the codes of conduct propose, and 18% of swimmers continued to touch and obstruct the whales. While the researchers in this study also conducted an extensive examination of codes of conduct published by zoos, we note that most guidelines have statements prohibiting visitors' intimate interaction with animals. The Panda Base Guidelines further exemplify the need for an animal-based organization to draw boundaries between animals and humans as a tool to supposedly limit impacts on the latter. We assert that the study's outcomes raise numerous questions that extend beyond immediate resolution. Within this context, our objective is to explore a specific inquiry: understanding why and how a code of conduct designed to foster an ethical relationship between animals and humans is less effective than its peripheral goals, such as discouraging smoking and preventing lawn trampling.

Intimate contact between humans and animals becomes a contested ethical ground in zoos. On the one hand, animal organizations use codes of conduct to prohibit such contact between species. On the other hand, visitors continue to seek ways to transgress these boundaries. For Foucault, the capitalist society builds on an imbalance between forces of taboo and transgression [76]. The violation of the animal-human boundary depends on the limitations that zoos establish, playing a constitutive role in general zoo experiences. According to McNay [76], transgression of the taboo "constitutes the necessary basis of human social life" and affirms the personal identity that contemporary beings are often unconscious of. From Foucault's perspective, the code of conduct prohibiting visitors from having intimate contact with animals is an invitation to violate the rule. Understandably, studies [55,56,77,78] have reflected that touching or intimate contact with animals can have a lasting and impressive impact on tourists. From Foucault's perspective, this lasting impression also results from violation of a code of conduct.

Foucault's insight, therefore, places the animal-human boundary at the centre of contemporary capitalistic society in which zoos exist. This constitutional role of transgressing the human-animal boundary makes the practice an inevitability for zoos—prohibiting visitors from having contact with animals in zoos constructs a taboo that necessarily entails its violation. The Panda Base Visitor Guidelines explicitly tell tourists that they are not allowed to have intimate contact with animals. However, we believe that the key message here is that the possibility of constructing an animal-human no-touch taboo juxtaposes an open invitation to actively seek to touch the animals. In fact, following Foucault's philosophy, an

ever-tightening boundary generates a greater desire and need for transgression with no alternative but the inevitable deconstruction of social structures.

In contrast, Kant's work provides an alternative to establishing code of ethics effectiveness. Regarding Kant's views on "the thing itself", we see a different perspective assembled around aesthetics, the beautiful and sublime in nature, as well as the belief that animals ought not to be used as a means to our selfish ends in view of their role as objects of pleasure and profit. Animals in Kant's view are organized beings with a purpose in the sense that they display "inner objective material purposiveness" [79]. Kant had bounded views on the degree to which we should extend moral consideration to animals. He argued that because they are not rational, that they should receive relative moral value rather than inherent value. We, therefore, have only indirect duties to animals rather than direct duties [80]. Rather than perceiving the no-touch policy in zoos as a taboo invoking its further violation, Kant's philosophy suggests that planning a more effective code of conduct in zoos needs the cultivation of respect towards animals.

Notwithstanding the limitations of the code of conduct, and following the lead of Kant, there is an argument that animals at the Panda Base ought to be extended respect through an ethic of care [34] as ends in themselves, i.e., animals should not be means to our selfish ends in a culture based on profit and pleasure. The way in which to move this agenda forward, we contend, is through education. Research indicates that rather than telling visitors NOT to touch or have intimate contact with animals (the deontological approach), explanations, and justifications can help build mutual respect and understanding between species through interpretive messages (the teleological approach). As Malloy and Fennell [22] have observed, the teleological instead of the deontological approach, helps the communication between codes of conduct and the participant. For Kant, the animal-human relationship does not necessarily build around a "No touching" policy in codes of conduct but asks for respect for animals' unique existence in zoos.

In their study on tourists' support for conservation messages and sustainable management practices in wildlife tourism, Ballantyne, Packer, and Hughes [81] (p. 658) suggest that animal-caring organizations could "enlist tourists as conservation partners", a process which builds on the communication of "reasons behind any constraints imposed" on wildlife. Seeing tourists as partners in conservation efforts urges the use of a teleological approach actuated through the code of conduct in zoos. For the Panda Base to foster a partnership with tourists, the current deontological approach to the code of conduct requires a change that will enable transparency in its communication and interaction with tourists.

## 6. Conclusions

Zoos worldwide have implemented the use of codes of conduct to regulate and manage the activities and actions of visitors. However, to our knowledge, few studies have considered the codes of conduct in zoos to be a legitimate field for research, even though zoos have been an ethical ground for negotiating the relationship between people and animals. Codes of conduct in zoos are uniquely placed, manifesting this ethical dialogue between species.

This study contributes to the under-researched area of code of conduct effectiveness in zoos by providing an overview of the readership of the codes of conduct, compliance with codes, and the ethical judgements by visitors at the Panda Base. General patterns emerged from the data. Panda Base visitors tend to skip the codes despite their belief in the necessity of the codes, and participants hold that intimate contact with animals is more ethically acceptable. At an attraction management level, managers should alter the Visitor Guidelines using a teleological approach to foster mutual respect and understanding between visitors and zoo animals. Furthermore, key messages of the Visitor Guidelines can be delivered through visual designs or graphs that can best attract attention from readers because a selective reading style is as effective as a thorough reading style. We suggest that further studies should investigate how visual designs may help improve the effectiveness

of the codes of conduct in zoos. Additionally, Foucauldian and Kantian philosophies were employed as theoretical frameworks to understand how zoos have facilitated the ethical relationship between visitors and animals. We contend that Kant's philosophy can provide better guidance in achieving a respectful relationship between animals and visitors in captive animal venues.

The present study is based on a survey of visitors to the Panda Base, which is limited to the Panda Base's uniqueness as an animal breeding organization. The giant panda is a celebrity animal [74], and its celebrity status marks a difference between the Panda Base and other zoological organizations. Also, this study uses the Visitor Guidelines of the Panda Base as its analytical field. We note the diverse and individualized approach that zoos employ when drafting their codes of conduct. The findings of the study raise further questions. For example, while demographic differences such as gender and age can potentially have a significant influence on the perceived importance of Guidelines, this study affirms the link between social media and the readership of codes, which should be analyzed in further research. In addition, the perceived importance of the Guidelines does not positively relate to the ethical soundness of conduct. In contrast, we observed that participants who claimed that the Guidelines were unnecessary suggested a lower acceptance of misconduct. Qualitative research can also benefit this field of study for the construction of a more effective and ethically grounded code of conduct.

The sample of this survey was limited to tourists visiting the Panda Base, decreasing the generalizability of the research results. However, the findings of the study are relevant to zoos that facilitate codes of conduct as a management tool for tourists. Additionally, the study only approached Chinese tourists whose cultural background could mean a significant difference in interpreting and reading the code of conduct at the Panda Base. Cross-cultural comparisons are needed to understand the dynamic inherent to cultural practices such as interpreting and facilitating a code of conduct. Considering that studies on visitor codes of conduct in zoos are limited, especially in China, the data presented in this study provide a starting point for researchers to conduct similar studies. The aim of all of these efforts should be a code of ethics that advances the wellbeing of animals foremost, and the ethical conduct of visitors to support such aim.

**Author Contributions:** Conceptualization, Y.G. and D.F.; methodology, Y.G.; software, Y.G.; validation, Y.G.; formal analysis, Y.G. and D.F.; investigation, Y.G.; resources, Y.G.; data curation, Y.G.; writing—original draft preparation, Y.G.; writing—review and editing, D.F.; visualization, Y.G.; supervision, D.F.; project administration, Y.G.; funding acquisition, Y.G. All authors have read and agreed to the published version of the manuscript.

**Funding:** This research received no external funding.

**Institutional Review Board Statement:** The study was conducted in accordance with the Declaration of Helsinki. Ethical review and approval were waived for this study due to the anonymous participation.

**Informed Consent Statement:** Informed consent was obtained from all subjects involved in the study.

**Data Availability Statement:** Please access the dataset of this study at: https://doi.org/10.6084/m9.figshare.25124564.v1.

**Acknowledgments:** We thank Qiaolin Chen for her consistent support.

**Conflicts of Interest:** The authors declare no conflicts of interest.

### Appendix A

Chengdu Research Base of Giant Panda Breeding Visitor Guidelines (Version of May 2023).

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
