# Peer review of "Codes of Conduct at Zoos: A Case Study of the Chengdu Research Base of Giant Panda Breeding"

_tourismhosp, doi:10.3390/tourhosp5010007_

Round 1
Reviewer 1 Report
Comments and Suggestions for Authors
This manuscript provides an overview of a study focused on the application of a code of conduct at Panda Base in Chengdu China. It provides a valuable examination of the role of codes of conduct in the context of animal-human interactions in a tourism space (i.e. zoo). However, some modifications are needed to improve the quality of the manuscript
Much of the Introduction is not clear. Please clarify on page 1.
Related to "However, studies consistently indicate that entertainment is the most important objective at zoos..." do visitors see the primary purpose of zoos as entertainment?
Connection between perceptions of entertainment and the "visitor effect" issue is not clear.
A clarified illustration of the problem being addressed (maybe it is animal welfare?) is needed at the end of the first paragraph
Pg. 2
The authors need to more clearly tie this work to tourism/hospitality and the journal's aims. The paragraphs at the top of page 2 feel like a logical place to do that.
What is meant by "the nature of zoo codes of conduct"
pg. 5
The first full paragraph on pg 5 does a good job of summarizing the literature review but could be strenghtened by reintegrating why zoos make sense as a focus area for examinations of codes of conduct
Pg. 6
Greater rationale on the three scenarios selected for study is needed.
Please clarify how this study is different from Fennell and Malloy (1999) in asking whether tourists misbehaved before taking the survey.
With the multiple options provided for section A in each scenario, could respondents select only one option? What if they did it AND they saw someone else doing it? This may be worth clarifying/describing for the purposes of replication of this work.
Pg. 13
Both the Discussion and Conclusion fail to fully integrate the impact this work has on tourism or hospitality management (they need to better align with the aims of this journal)
Pg. 15
Limitations should also address any potential issues related to social response bias
Reviewer 2 Report
Comments and Suggestions for Authors
Congratulation to your work! I find the topic of submitted manuscript highly intriguing. Methods as well as results are clearly described. The literature review in this research has been meticulously composed, offering a comprehensive framework that substantiates the study. The Discussion and Conclusion sections of the manuscript are commendable. The authors effectively synthesize the findings, contributing valuable insights to the field. Overall, an insightful and well-structured conclusion enhances the quality of the research.
I have just minor suggestions for the authors:
The map of the location of Chengdu Panda Base is absent.
The line numbers are missing in the paper ( for reviewers it is easier writing a review with line numbers).
In some paragraphs of the paper, numbers are inconsistently presented, alternating between words and numerals. Please ensure uniformity in this regard. For example check the paragraphs in section '4. Results, 4.1. General summary'.
Comments on the Quality of English LanguageI would rewrite the beginning of last paragraph (Section 3.1. Panda Base’s “Visitor Guideline”): 'Based on our review of literature review,.....' to 'Based on our literature review, .....'
Reviewer 3 Report
Comments and Suggestions for Authors
The contribution proposes an interesting study which can be of interest for the reader and the scientific community. It needs minor revisions in order to be suitable for publication.
Below the revisions that are suggested. Please check references in the text, especially in the Introduction section, and revise them in accordance with the standards of the journal
The absence of Line numbers makes more challenging to be precise in the suggestions provided.
Abstract
1. “provide an opportunity to consider the significance of ethical codes in zoos”: see below when codes of conduct are distinguished from codes of ethics. Can you clarify or reformulate?
Introduction
The reader can feel confused about the difference between codes of conduct and codes of ethics. Please clarify and uniform
-“However, studies consistently indicate that entertainment is the most important objective of zoos”: this sentence is questionable, see recent IUCN statement for instance. It is suggested to add ‘one of the most…’
-“These effects are taking placing": place
-“often articulated within the broader context of compassionate conservation”: this sentence is questionable: can you better explain and support it?
- “codes of ethics in tourism”: codes of ethics or codes of conduct?
- “codes of conduct in zoos have come to embody this relationship”: codes of conduct or codes of ethics? In general this sentence is rahter obscure. Can you clarify it?
- “This competing demands…”: it is suggested to delete 'this'
2. Literature review
It is suggested to justify the need for a Literature review to support what was presented in the Introduction, by adding an explanation of what folllows in the Introduction itself.
2.1.
It is also suggested to introduce the following section, by guiding the reader through the structure of the argumentation, otherwise the reader may feel disoriented
2.2.
“For these authors, the most effective codes are teleological rather than deontological because they provide the rationale and justifica-tion behind the use of codes”: can you better explain this point?
“Stemming from the World Association of Zoos and Aquarium’s (WAZA) [53] developed code of ethics and animal welfare, as well as the World Zoo and Aquarium Animal Wel-fare Strategy [54], and WAZA resolution on animal interactions [55], WAZA [56] devel-oped “Guidelines for Animal-Visitor Interactions”: just for clarity, please reformulate this sentence by first mentioning WAZA and then mentioning the different documents
3. Method
3.1.
“Based on our review of literature review”: although the input is very interesting, it has to be better explained and justified in the relevant paragraph and paired with the description of the investigation that follows
3.2.
“This paper replicates the Multidimensional Ethics Scale (MES) Fennell and Malloy [60] adopted from Reidenbach and Robin”: why the MES, can you better justify this choice?
4. Discussion
It is suggested to revise the paragraphs dedicated to Kant in order to better highlight the link with the results of the survey, especially the results of the application of the MES.
Moreover, the last paragraph of the Discussion section is reproposing the interesting input provided in section 2.1 and 3.1. it is suggested to better highlight the link with the results of the survey of this input also.
Conclusion
- “Additionally, Foucauldian and Kantian philosophies were employed as theoretical frameworks to understand how zoos have facilitated the ethical relationship between visitors and animals”: this point should be clarified since the beginning, in the Introduction and 2.1. sections, in order to guide readers to better understand the link between the two parallel investigations developed in the contribution.
- “The present study…of misconduct”: it is suggested to move this paragraph to the Discussion section of the contribution
Reviewer 4 Report
Comments and Suggestions for Authors
First of all, I would like to congratulate authors for this interesting paper.
In the introduction section, authors discuss that Zoo codes of conduct must specify visitor obligations and responsibilities in order to achieve conservation and education objectives. Therefore, this research aims to investigate the nature of zoo codes of conduct in order to establish a fundamental understanding of their use, ethical value, and compliance among visitors.
However, in page 5, authors list the focus of their investigation in detail since they state that they aim to investigate (1) visitors' acceptance of the Visitor Guideline, (2) whether tourists noticed the Visi[1]tor Guideline when booking online and how they interacted with the guideline window, and (3) tourists’ evaluation of the importance of the Visitor Guideline.
I would advise to rewrite the introduction’s aims with this detail because it helps readers to better understand this interesting discussion.
The literature review is quite good and based on quite a good number of references. These balance older with more recent references and this is important to understand the theoretical evolution of this study. Anyway, there could be some more recent references in this section.
This study is based in a sample of 849 visitors which is quite good. The statistical discussion helps to achieve the conclusions and understand all of the main issues concerning this paper.
REferences need some minor revision according to ther Journal's indications.
Round 2
Reviewer 1 Report
Comments and Suggestions for Authors
The authors have sufficiently addressed the reviewers' concerns.
Reviewer 2 Report
Comments and Suggestions for Authors
Thank you for including all of my comments and suggestions to your manuscript. Congratulations to great work and good luck with your next research.
Reviewer 4 Report
Comments and Suggestions for Authors
Congratulations on this final version.
You have successfully changed the paper and this final version is quite good and appropriate for the current journal.
You have corrected the introduction and this version is quite clear and objective.
The literature review has now an important introduction and some minor corrections that improved its consistency.
Final discussion is now corrected, and the corrections made by the authors allow readers to better understand its main aspects.
Conclusions were also rewritten and this version allow readers to fully understand its impacts and challenges which is quite important.
Finally, final references were corrected according to APA norms.